# A Systematic Literature Review of Mitochondrial DNA Analysis for Horse Genetic Diversity

**DOI:** 10.3390/ani15060885

**Published:** 2025-03-20

**Authors:** Ayman Agbani, Oumaima Aminou, Mohamed Machmoum, Agnes Germot, Bouabid Badaoui, Daniel Petit, Mohammed Piro

**Affiliations:** 1Laboratory of Veterinary Genetic Analysis, Department of Medicine, Surgery and Reproduction Agronomic, Veterinary Institute Hassan II Rabat, Rabat B.P. 6202, Morocco; oumaimaminou18@gmail.com (O.A.); mm.machmoum@gmail.com (M.M.); vetpiro@yahoo.fr (M.P.); 2LABCiS, UR 22722, Faculty of Sciences and Technology, University of Limoges, F-87000 Limoges, France; agnes.germot@unilim.fr; 3Department of Biology, Faculty of Science, Mohammed V University, Rabat B.P. 8007, Morocco; bouabidbadaoui@gmail.com

**Keywords:** mitochondrial DNA, equine maternal lineages, horse genetic diversity, mtDNA haplogroups, phylogeny, population genetics, conservation genetics

## Abstract

For effective conservation and breeding programs, investigating horse genetic diversity is essential. This study reviews 76 studies published since 2012 on mitochondrial DNA (mtDNA) analysis, with a focus on maternal lineages and haplogroups. The findings highlight persistent challenges, particularly the need to update the 2012 haplogroup classification system and mitigate data loss caused by sequence trimming. A more comprehensive understanding of equine genetic diversity requires addressing underrepresented breeds and regions, which remain overlooked in existing studies. Additionally, integrating mtDNA findings with performance traits and whole-genome sequencing would enhance genetic evaluation, providing valuable insights for conservation and management. Ultimately, these advancements are crucial for ensuring the sustainable preservation of horse populations worldwide.

## 1. Introduction

Horses have played a significant role in human culture and history, making the study of the origins of their domestication particularly interesting. Archaeological evidence suggests that the earliest domestication of horses occurred in the Eurasian steppes between the fifth and fourth millennia BC [1]. Understanding the genetic diversity of horse breeds is crucial for the conservation of endangered and declining populations [2]. mtDNA analysis is a very informative tool in this endeavour, playing a key role in population genetics and molecular phylogenetics [3].

mtDNA possesses unique properties: it is maternally inherited, is haploid, is non-recombining, and has a higher mutation rate than nuclear DNA [3]. These characteristics enable the identification of intra- and interbreed relationships [4,5,6], especially when combined with historical data [7]. mtDNA has been extensively used in phylogenetic studies of domestic horses, providing valuable clues to their origins and tracing ancestry through maternal lines [8]. Moreover, mtDNA analysis can be used to verify the reliability of maternal lineages recorded in studbooks [3]. The analysis of various breeds has also revealed different clusters, called haplogroups, and their geographical patterns, with several geographical centres characterised by high frequencies of certain haplogroups [9,10,11].

This Systematic Literature Review (SLR) synthesises existing research on equine mtDNA analysis. It includes studies published since 2012 and retrieved from five major databases: Scopus, PubMed, Research4Life, Web of Science, and ScienceDirect. The main objectives of this review are to examine the methodologies used for mtDNA analysis, to compare findings between studies, and to identify trends or gaps in the research.

## 2. Materials and Methods

### 2.1. Methodology

This systematic literature review followed the PRISMA methodology (Preferred Reporting Items for Systematic reviews and Meta-Analyses) [12]. Two researchers (AA and OA) participated in the extraction and filtering stages of this study. This study began with a broad search across five databases: Scopus, PubMed, Research4Life, Web of Science, and ScienceDirect. The search used a combination of keywords and logical operators, such as AND, OR, and NOT. The exact keyword logic applied was the following: (Horse OR Equine) AND (“Mitochondrial DNA” OR “mtDNA” OR “Mitochondrial Genome” OR “D-loop” OR “Maternal genetic diversity”). The extraction was performed on 22 July 2023. For the screening process, we used Zotero to manage the references and remove duplicates. We excluded grey literature, such as conference proceedings, articles published in journals without an SJR index, as well as articles published before 2012. This cut-off was chosen because 2012 marks the publication of Achilli et al.’s [9] classification system, which remains the most recent and comprehensive framework for equine mtDNA haplogroup classification. To ensure consistency, both researchers independently screened and assessed the articles. Any discrepancies were resolved through discussion and consensus.

### 2.2. Inclusion and Exclusion Criteria

We included studies that focused on the genetic diversity of horses through the mtDNA and were published in peer-reviewed journals from 2012 onwards. Studies were excluded if they did not meet these criteria or if an English version was unavailable.

### 2.3. Eligibility and Screening

During the eligibility step, we first read the abstracts and excluded papers that did not focus on mtDNA analysis or were unrelated to horses. We then conducted a full-text assessment, retaining only the studies that applied mtDNA analysis to equine genetics. The included papers were further sorted to identify those studying the genetic diversity of horses.

### 2.4. Update

The list of articles concerning the genetic diversity of horses using mtDNA was updated on 28 November 2024, adding 5 articles to the list. The same search keywords were applied, restricting results to publications between 23 July 2023 and 28 November 2024. The titles were screened first, followed by the abstracts, and relevant articles were incorporated into our analysis.

### 2.5. Data Extraction Process

The selected papers were carefully analysed to extract the sampling methods (type and size), the mtDNA region studied, the method of sequencing, the bioinformatics and statistical tools and techniques employed, the metrics used for evaluation, and the haplogroups identified. To ensure consistency and completeness, we systematically extracted the same set of variables from each study (Appendix A).

### 2.6. Data Synthesis

The data from the included studies were synthesised both qualitatively and quantitatively to provide a comprehensive overview of the genetic characterisation of horses using mtDNA analysis. We created a detailed table (Appendix A) to extract and organise the information, including the following:Date of publication;Breed studied;mtDNA applications;Additional methods used in addition to mtDNA sequencing;Sample type and size;Number of GenBank sequences;mtDNA region studied;Number of base pairs (bp);Position of nucleotides in the mtDNA genome;Whether the study was trimmed or not;Sequencing technique;Tools used for the analysis;Metrics and methods for evaluation;Country of the study;Haplogroups found.

The different figures were created using Python 3 (in Jupiter Notebook 7.0.8), which facilitated the visualisation of the data. Additionally, we attempted to calculate the correlation between different variables to explore potential links.

To compare mitochondrial haplogroup classifications (Achilli, Cieslak, and Jansen), we aligned representative sequences from each system. Some Jansen haplogroups were unavailable, so we selected accessible sequences and trimmed them to 386 base pairs for consistency. Phylogenetic trees were constructed with MEGA version 12 [13] using Minimum Evolution (500 bootstrap iterations) and Maximum Likelihood (Tamura-Nei model, 500 bootstrap iterations). The Tamura-Nei was provided by the program implemented in MEGA version 12. These analyses produced a correspondence table linking haplogroups across classifications.

## 3. Results

A total of 1380 articles were initially detected when we extracted the articles from the five selected databases. After the screening step, 350 articles remained. Following the eligibility step, we finally included 71 articles in our review (Figure 1). An additional five articles were found in the update of 28 November 2024 [3,5,6,9,14,15,16,17,18,19,20,21,22,23,24,25,26,27,28,29,30,31,32,33,34,35,36,37,38,39,40,41,42,43,44,45,46,47,48,49,50,51,52,53,54,55,56,57,58,59,60,61,62,63,64,65,66,67,68,69,70,71,72,73,74,75,76,77,78,79,80,81,82,83,84,85].

Most of the 76 studies follow a consistent pipeline from the selection of samples to the interpretation of results. The exact steps are as follows: sampling, DNA extraction, PCR amplification, sequencing, sequence alignment, phylogenetic analysis, statistical analysis, and interpretation. Some articles deviated from the standard pipeline. For example, Glazewska et al. [25] started directly with phylogenetic analysis using only public data from GenBank, while Sharif et al. [78] did not use sequencing techniques but instead used genotyping with SNP variants across the entire length of the mtDNA.

To explore the content of the articles, several approaches were conducted. First, no clear trend in the topics was observed when examining the publication dates of the articles; the studies were published consistently across the years without any significant peaks (Appendix A). Second, we created a correlation matrix between various quantitative variables, including sample size (the number of chosen samples to be sequenced), GenBank sequences (additional samples retrieved from the GenBank database), total analysed sequences (the sum of the sample size and GenBank sequences), mtDNA number of base pairs (bp) sequenced, and base pairs remaining after trimming sequences for analysis. No meaningful correlation was found between the variables.

The included studies demonstrated various applications of mtDNA analysis (Appendix A). The majority (85.5%) used mtDNA to analyse the population structure of a given group. Almost all studies (97.3%) applied mtDNA to compare phylogenetic relationships, either between specific populations or in relation to different haplogroup systems, as well as to determine the origin of a breed. Additionally, 48.6% of the studies discussed conservation strategies based on mtDNA findings, while 11.8% used mtDNA to verify studbooks and pedigree records of their horses (Figure 2).

The key findings from these studies converge on several themes, with some variations. Most research focuses on the genetic diversity of specific horse breeds, often confirming historical records [21,46], hypothesising about their origins [15,18,80], or exploring their genetic relationships with other breeds [31,32,53,54]. For instance, Giontella et al. [15] supports the hypothesis that both humans and Giara horses migrated from the Eastern Mediterranean during the first millennium BCE.

Beyond origins, several studies document the loss of genetic diversity over time, particularly the decline of specific haplotypes [24,27,79]. In Chinese indigenous horses, haplogroups B, F, and G have decreased in frequency in recent years [24]. Similarly, Spanish endangered breeds have lost maternal lines [79]. This decline highlights the connection between genetic diversity loss and breed extinction, reinforcing the need to integrate mtDNA with autosomal and paternal markers for more effective conservation strategies [24].

Selection pressures have also shaped horse populations. Novoa-Bravo et al. [60] demonstrated microevolutionary changes leading to the divergence of two breeds due to selective breeding. Winton et al. [23] emphasised the role of subpopulation analysis in understanding the genetic impact of past management practices and guiding future conservation efforts.

Finally, in pedigree verification, some studies confirm the reliability of studbooks [83]. Bower et al. [82] reports that maternal sub-lineage records for Thoroughbred racehorses are 92% accurate compared to only 60% for general maternal lineage records. However, other studies reveal inconsistencies in pedigree documentation [20,72,79], underscoring the need for genetic tools to enhance accuracy in pedigree management.

Although the majority of the articles focused solely on mtDNA, a variety of other genetic markers were incorporated with mtDNA analysis. The most frequent were microsatellites and the Y chromosome, while pedigree analysis, Single Nucleotide Polymorphisms (SNPs), specific genes (such as the gait and coat colour genes), and even DNA fragments were rarely associated (Figure 3).

### 3.1. Sampling

For the first sampling step, we considered two parameters: the type and the size of the sample. We also included sequences extracted from GenBank and used in molecular comparative analyses to increase the overall sample size. Among the studies that reported sample type as sources of mtDNA, 53% used blood, 32% used hair, and 13% used both (Figure 4). Two studies within our review employed less conventional sampling methods: One study used buccal swabs in conjunction with blood and hair samples [57], while another successfully extracted DNA from faecal matter [61].

In two studies, no new samples were collected; instead, data were sourced directly from GenBank. These studies relied on previously established haplotypes and their affiliations to dam lines, which had been deposited in GenBank from earlier research. The two articles that did not use new samples (Appendix A) adopted this indirect approach, leveraging the GenBank data to analyse haplotype distributions [25,53].

Regarding sample size, the minimum and the maximum were 2 and 1582, respectively, and the median was 110 (Figure 5a). Including sequences obtained from GenBank, the minimum and the maximum increased to 17 and 3965, respectively, and the median increased to 192 (Figure 5b). Analysing the total number of samples over time by sample type revealed no discernible pattern or trend, suggesting that sample type and size were determined by the specifics of each study.

### 3.2. DNA Extraction

DNA extraction follows various standard protocols to obtain high-quality DNA. For example, the Illustra blood genomic Prep mini spin kits (GE Healthcare, Little Chalfont, UK) [14] or protocols described by Miller [86] were employed [29,31]. The choice of protocol depends on the specifics of each study and the laboratory conditions.

### 3.3. PCR Amplification

We extracted data on the mtDNA region, number of sequenced base pairs, and nucleotide range from each article. The partial D-loop region was analysed in 58 articles, the whole D-loop region in 7, the entire mtDNA in 4, partial mtDNA in 3, Cytochrome b gene (*CYTB*) in conjunction with the partial D-loop region in 2, and *ATP6* gene and SNP variants in mtDNA in 1 case each (Figure 6). For partial D-loop studies, the number of base pairs ranged from 202 to 752, with a median of 476. Approximately 36% of the articles trimmed the sequences to use GenBank references for detailed analysis. For trimmed sequences, the base pair count ranged from 206 to 608, with a median of 312.

Among the studies that specified the positions of amplification, the reported ranges fell between positions 15,341 and 16,642 of the referenced horse mtDNA, with all of them overlapping in the 15,579–15,740 portion (Figure 7).

### 3.4. Sequencing Techniques

Regarding the sequencing techniques, all included studies used Sanger sequencing, which remains the standard for smaller-scale studies and, particularly, for mtDNA analysis due to its reliability, cost-effectiveness, and accuracy in targeted regions. Instruments such as the 3130 Genetic Analyzer (Applied Biosystems, Foster City, CA, USA) and the 3730xl 96-capillary DNA Analyzer (Applied Biosystems Inc., Foster City, CA, USA) were commonly used. None of the included studies utilised Next-Generation Sequencing (NGS) for the characterisation of genetic diversity. However, NGS was applied in related studies to sequence the complete mitogenome of specific horses, generating reference sequences rather than directly analysing population-level genetic diversity [87].

### 3.5. Analysis Tools

The tools used in the studies were categorised into three main groups: (i) bioinformatics tools, (ii) phylogenetic analysis tools, and (iii) population genetics and statistical analysis tools. Multiple tools were employed across the studies, with some being more frequently used. The most commonly used tools included DnaSP for DNA sequence polymorphism analysis [88], MEGA for phylogenetic analysis [89], Network for phylogenetic network construction [90], Arlequin for population genetics analysis [91], Clustal for multiple sequence alignment [92], BioEdit for sequence editing [93], and BLAST for sequence local alignment [94] (Figure 8). These tools are widely adopted due to their functionality, ease of use, and integration into established workflows.

### 3.6. Metrics

A wide range of metrics were used. Among the most common are ‘number of haplotypes’ used in 93% of the articles, ‘haplotype diversity’ used in 66% of the articles, ‘nucleotide diversity’ used in 66% of the articles, ‘Median joining (MJ)’ used in 58%, ‘polymorphic sites’ used in 56%, ‘Neighbor Joining’ used in 43%, pairwise *F*_ST_ (Fixation Index) used in 34%, and ‘average number of nucleotide differences’ used in 20% (Figure 9). The MJ model is frequently used despite criticism of its use in inferring evolutionary relationships [78,95].

The metrics used in the studies can be grouped into four categories. The Genetic Diversity and Population Structure Metrics category (used in 70 articles) includes measures like the number of haplotypes, haplotype and nucleotide diversity, and gene diversity. The Phylogenetic and Evolutionary Analysis category (63 articles) features methods such as Neighbor Joining, Median Joining, Maximum Likelihood, and Bayesian approaches. The Selection, Neutrality, and Demographic Tests category (16 articles) includes Tajima’s D, Fu’Fs and Li’s D tests, Harpending’s Raggedness Index, the cross-population EHH test, mismatch distribution, and the stepwise mutation model. Finally, the Statistical and Multivariate Analysis category (13 articles) covers PCA (Principal Component Analysis), PCoA (Principal Coordinates Analysis), cluster analysis, DAPC (Discriminant Analysis of Principle Components), ANOSIM (Analysis of Similarities), Chi-square tests, and Φ-statistics.

### 3.7. Haplogroups

At the end of the analysis, some articles simply reported the number of haplotypes found, while others classified their haplotypes into haplogroups. Despite nearly half the articles adopting the 2012 classification system by Achilli et al. Ac [9], various other systems are still in use (Figure 10).

Vilà et al. [96] conducted one of the earliest studies to attempt such classification, identifying six divergent clades (A to F) based on 355 bp from the left domain of the D-loop region (Table 1). Their analysis included 191 horses from 10 breeds as well as additional sequences retrieved from GenBank. The study emphasised the high mtDNA diversity in horses and suggested widespread incorporation of wild matrilines into domestic populations. Jansen et al. [11] expanded on this by classifying haplotypes into 17 distinct phylogenetic clusters (A1 to A6, B1, B2, C1, C2, D1, D2, D3, E, F1, F2, G). This classification was based on 469 bp of the D-loop region sequenced from 652 horses across 25 breeds, using a reduced median network approach. Cieslak et al. [10] proposed a different system, identifying 19 haplogroups (A to K and X1 to X7) by analysing both the D-loop (722 bp) and HVR1 regions (247 bp) in 1961 sequences, including 207 ancient remains and 1754 modern horses from 46 primitive breeds. Their analysis relied on the ML method and introduced a new haplogroup nomenclature to address the high variability in horse mtDNA. Another set of studies adopted the system proposed by Achilli et al. [9], who took a more comprehensive approach by sequencing the complete mitochondrial genome (~16,500 bp) of 83 horses from 27 breeds. They classified sequences into 18 haplogroups (A to R) using the ML method. Achilli et al. [9] highlighted the importance of analysing complete mtDNA sequences to avoid biases and noted that relying solely on partial sequences, such as the D-loop or HVR1 regions, could overlook important phylogenetic relationships.

We generated phylogenetic trees for the three main classification systems (Achilli, Cieslak, and Jansen) using Achilli’s system as the reference (Appendix A). These trees allowed for us to establish a correspondence table (Table 2) aligning haplogroups from Cieslak and Jansen with those of Achilli.

For most haplogroups, the correspondence was consistent between the two phylogenetic methods used—Minimum Evolution (ME) and Maximum Likelihood (ML). However, some discrepancies were observed; for instance, Cieslak’s K3b corresponded to haplogroup O in ML but to P in ME. Several haplogroups from Jansen and Cieslak had no equivalent in Achilli’s classification. Notably, A1, identified in Jansen, lacked an assigned cluster in Achilli’s system. Cieslak’s classification also included extinct haplogroups, such as X3c1a, X6c, X8a, X13, X14, G2, and B3. Some of these haplogroups (G2, B3, X14, X6c, and X8a) had correspondences in one or both phylogenetic techniques, while others (X3c1a and X13) had no equivalent in Achilli’s classification.

Additionally, certain haplogroups still present in modern horses, such as A, K, K1, and X3c1, remained unclassified in Achilli’s system in at least one method (K1 and A in one, and K and X3c1 in both). Moreover, our review highlighted that several studies have reported haplotypes that do not fit into any of the existing classification systems [39,40,45,81].

Some articles excluded strong mutational hotspots, specifically the 15,585; 15,597; 15,604; and 15,650 positions of the horse mtDNA [36,81,85], while others excluded certain hotspots, such as 15,585; 15,597; and 15,650; and down-weighted others, such as 15,659 and 15,737 [80]. In addition, some studies down-weighted mutational hotspots to 0.5 [20], and others double-weighted positions such as 15,532 [81]. Several articles did not mention these mutational hotspots at all.

### 3.8. Global Distribution of Studies

Since 2012, studies on maternal genetic diversity using mtDNA have been heavily concentrated in specific countries (Appendix A). China leads with 11 studies, and Italy follows with 9, reflecting their active research efforts. However, vast regions like Sub-Saharan Africa, Oceania, and parts of South America remain underrepresented, creating significant gaps in the global understanding of maternal genetic diversity (Appendix A). Given that each country often has multiple breeds, addressing these gaps will require multiple studies to achieve comprehensive coverage of horse populations in underrepresented regions.

### 3.9. Breeds Studied

In terms of breeds, disparities are also evident. The Arabian horse has been extensively studied, appearing in 10 studies, while many other breeds are mentioned only once (Appendix A).

## 4. Discussion

Despite its advantages, mtDNA alone provides a limited view of genetic diversity as it traces only the maternal lineage. This means it does not offer direct information about paternal ancestry or gene flow through stallions. Therefore, a more comprehensive approach would involve combining mtDNA analysis with other genetic markers, such as the Y chromosome or autosomal markers, to provide a broader perspective. While mtDNA has some limitations in establishing clear links between mtDNA haplotypes, breeds, and geographical distribution, the paternally inherited Y chromosome complements this by providing a robust genetic marker for investigating the origin and influence of recent paternal lineages [78]. Integrating both maternal and paternal markers could enhance the accuracy of genetic studies, particularly in resolving uncertainties related to breed classification and evolutionary history.

Most of the articles followed a clear methodological pipeline, yet significant differences were observed in the specific approaches used at each step. No correlation was found between the key variables extracted, suggesting that methodological choices were shaped by diverse factors. For instance, sample sizes are influenced by the size of the existing population, the access to the horses or their biological material, and the specific objectives of each study—whether aiming for broad comparative analyses or focusing on a single population. Similarly, the number of base pairs sequenced varied, potentially influenced by financial constraints, researcher preferences, or the need to align with available reference sequences in GenBank.

Blood was the most commonly used sample because it provides sufficient high-quality DNA for genetic analysis, though collection can sometimes be challenging. Non-invasive sampling methods —including faeces, hair, and saliva—are easier to collect and less invasive than blood. As a result, they are increasingly used to address animal welfare concerns [97]. In a systematic literature review, Zemanova et al. [97] reported a steady increase in the use of non-invasive methods in wildlife research.

Comparative studies have evaluated DNA isolation efficiency across different sample types. Gurău et al. [98] compared DNA extraction from blood and hair follicle samples in goats, finding that, while whole blood yielded slightly higher DNA quantities, both sample types provided sufficient DNA for amplifying 1200 bp fragments. Similarly, Muko et al. [99] assessed DNA yield and quality from oral mucosa swabs and faeces in horses, comparing them with DNA from blood. Although non-invasive samples provided sufficient DNA for PCR analysis, contamination from microorganisms and feed remained a concern.

In a review of 113 studies, Zemanova et al. [97] found that 94% of studies reported equivalent or superior performance of non-invasive genetic assessments compared to invasive sampling. Non-invasive techniques can also be more cost-effective and time-efficient. Thus, adopting non-invasive extraction methods could simplify DNA collection while maintaining comparable DNA quality. This approach may also achieve similar sequencing performance for the entire D-loop region of mtDNA, which is approximately 1192 bp long [46].

Sample sizes varied widely, with some studies focusing on nearly extinct breeds with fewer than 20 individuals, while others aimed to assess the global distribution of horses, incorporating thousands of sequences into their analyses. Smaller sample sizes may limit the ability to detect rare haplotypes, potentially underestimating genetic diversity within a population. Conversely, larger sample sizes generally enhance the statistical power of analyses but may introduce biases if certain breeds or regions are disproportionately represented.

The choice of the mtDNA region to be analysed and the selection of base pairs and nucleotide ranges influence the primers used during PCR amplification. The horse mtDNA is 16,660 bp long and contains 13 intron-less protein-coding genes, 22 transfer RNAs, two ribosomal RNAs, and the control region [100,101]. The D-loop, cytochrome b, *12S rRNA*, and *16S rRNA* can be used for species identification [66]. Most studies focused on the D-loop, which is widely used due to its high mutation rate and informativeness for maternal lineage studies. However, only two studies analysed cytochrome b, and both of these also included the D-loop [41,47]. Sziszkosz et al. [47] compared the two markers and found that some haplotypes were exclusive to cytochrome b and others to the D-loop, while some appeared only when combining both markers. These findings suggest that using multiple mitochondrial markers provides more precise results than analysing the D-loop alone, as relying solely on one region may overlook critical genetic variations.

Another crucial factor is the number of base pairs sequenced and the impact of trimming. While sequencing a small portion of the D-loop is cheaper, this approach may limit the depth of genetic insights. A significant amount of data can be lost when sequences are trimmed for comparison, as shorter sequences may not capture the full genetic diversity present in the mtDNA. For example, Khanshour et al. [5] demonstrated that transitioning from partial to whole D-loop sequencing increased the number of detected haplotypes from 74 to 97, revealing additional genetic diversity with longer sequences. However, Ma et al. [80] showed that a haplogroup identified from the complete mitochondrial genome may correspond to one or more of its HVR1 (Hypervariable region 1) sequences. This pattern was further studied by Musial et al. [65], who reported twice as many detected haplotypes when analysing the whole D-loop compared to using only HVR1.

The selection of analytical methods in mtDNA studies is closely linked to research objectives and dataset characteristics. Different metrics and phylogenetic approaches are favoured based on their reliability, interpretability, and alignment with common genetic analyses. For example, haplotype diversity and nucleotide diversity are often employed to assess genetic diversity. Well-established and standardised metrics are preferred, such as *F*_ST_, which is commonly used to measure genetic differentiation between populations. Specific methods are chosen based on the research questions and hypotheses being tested. For example, Median-Joining Networks are frequently used to visualise haplotype relationships within populations, while Bayesian inference and Maximum Likelihood (ML) are commonly employed in phylogenetic studies to construct evolutionary trees and infer species relationships. Bayesian inference, particularly with programs like MrBayes [102], consistently outperforms other methods in terms of accuracy and computational efficiency [103], especially with sparse data or small sample sizes [104]. Nevertheless, Bayesian approaches can sometimes overestimate confidence levels, leading to inflated certainty in phylogenetic trees [105]. On the other hand, Neighbor-Joining (NJ) is a simpler and faster technique mainly used for exploratory purposes. It does not require extensive model testing and can swiftly deliver an initial overview of the dataset. However, NJ may be less dependable for intricate datasets when compared to ML methodologies. Although ML techniques are computationally demanding, they evaluate multiple evolutionary models to determine the most appropriate one before constructing phylogenetic trees. This process typically yields more robust and accurate results. The choice of method depends on the specific context of the analysis, including the dataset size, complexity of the model, and the need for computational efficiency versus accuracy.

Our results showed that several non-extinct haplogroups from Cieslak and Jansen had no equivalent in Achilli’s classification, and some studies identified haplotypes that do not fit into any known system. This suggests that Achilli’s framework may be incomplete and that our broader understanding of equine mitochondrial lineages remains limited. The variety of haplogroup classification systems reflects ongoing efforts to characterise equine maternal lineages, yet their inconsistencies pose challenges for comparative studies. Differences in nomenclature, resolution, and reference datasets can impact phylogenetic accuracy, lineage tracing, and conservation planning. The absence of standardised criteria may lead to discrepancies in haplogroup assignment, limiting the reproducibility of findings across studies. Furthermore, the incompleteness of existing systems, as seen in our results, suggests that some mitochondrial lineages remain unclassified, which may lead to their loss from genetic records.

Additionally, geographical and breed representation in mtDNA studies remains uneven, limiting our understanding of global equine genetic diversity. The underrepresentation of certain regions underscores the need for more geographically diverse studies to achieve a comprehensive understanding of maternal genetic diversity. Expanding research efforts to Sub-Saharan Africa, Oceania, and South America would provide valuable insights into unique, yet overlooked, horse populations. To refine this understanding, it may also be beneficial to examine the global distribution of studies using genetic markers beyond mtDNA. This would help assess which horse populations have been studied and highlight gaps in broader genetic research. The concentration of studies on certain breeds, such as the Arabian horse, reveals an imbalance. Many breeds remain understudied, and additional research is necessary to fill these gaps.

## 5. Limitations and Perspectives

### 5.1. Limitations

Several limitations must be considered in interpreting our findings. First, in our comparison of haplogroup classification systems, sequences were trimmed to 386 base pairs to ensure consistency across datasets. This reduction may have led to information loss, potentially affecting haplogroup correspondence. Many studies also trimmed their sequences to align with GenBank references, a practice that can obscure genetic variation.

Another limitation is the reliance on mtDNA alone for genetic characterisation. While mtDNA is widely used for maternal lineage tracing, it does not capture paternal ancestry or autosomal variation, limiting the broader understanding of genetic diversity. Even within mtDNA-based studies, most rely on partial D-loop sequences, despite the availability of other informative mitochondrial markers, such as cytochrome b *(CYTB)* and protein-coding genes. This narrow focus reduces the accuracy of haplogroup assignments and overlooks phylogenetic relationships that could be better resolved with whole mitochondrial genomes.

Geographical and breed representation in equine mtDNA studies remains skewed, limiting the global understanding of mitochondrial diversity. Certain regions remain underrepresented, despite their potential to reveal unique and uncharacterised maternal lineages. Similarly, research has largely focused on a few well-known breeds, while many local and indigenous breeds remain unexamined. Addressing these gaps is crucial for obtaining a complete picture of equine maternal diversity.

The variety of haplogroup classification systems further complicates comparative studies. Several haplotypes identified in Cieslak and Jansen remain unclassified in Achilli, suggesting that the current classification frameworks are incomplete. Additionally, some studies reported haplotypes that do not fit into any established system, emphasising the need for a unified and expanded classification approach to improve consistency and reproducibility in equine mtDNA research.

### 5.2. Perspectives

Future studies should prioritise the use of additional genetic markers and longer mtDNA sequences to enhance the resolution of lineage characterisation. Expanding beyond partial D-loop sequencing to include *CYTB*, *COX1* gene, and potentially full mitochondrial genomes would improve haplogroup assignment and phylogenetic accuracy. Revisiting previously studied breeds using longer sequences could also minimise data loss from sequence trimming and improve genetic profiling.

Expanding research efforts to underrepresented breeds and regions is essential for a more comprehensive understanding of equine mitochondrial diversity. Increasing sampling from Africa, South America, and Asia could uncover previously undocumented haplotypes and refine existing classifications.

Ancient DNA (aDNA) research provides another critical avenue for mitochondrial lineage studies. Some extinct lineages identified in Cieslak and Jansen remain unclassified in Achilli, raising questions about how ancient sequences integrate into modern haplogroup frameworks. Future studies should compare ancient D-loop sequences from Cieslak and Jansen against Achilli’s dataset to determine which lineages can be classified and which remain unresolved. Given the limitations of short sequences for such comparisons, whole mitochondrial genome sequencing of ancient remains should be prioritised to provide a more robust dataset for phylogenetic analysis.

A major step toward improving mitochondrial studies is the standardisation and unification of haplogroup nomenclature. The discrepancies between Achilli, Cieslak, and Jansen highlight the need for an updated and expanded classification system that integrates both modern and ancient mitochondrial sequences. Expanding haplogroup reference databases by sequencing additional complete mitochondrial genomes, particularly from breeds reporting unclassified haplotypes, would refine classification accuracy and bridge the gaps between existing systems.

Finally, Next-Generation Sequencing (NGS) presents a powerful tool for addressing these research gaps. NGS generates significantly more data than Sanger sequencing in shorter timeframes, making it highly efficient for whole mitochondrial genome analysis. Its accuracy and cost-effectiveness allow for the simultaneous sequencing of multiple mtDNA genomes, providing deeper insights into genetic diversity and phylogenetic relationships. NGS is particularly valuable for degraded and ancient samples, as demonstrated in paleogenomic research [106]. Applying NGS to ancient horse remains could clarify missing lineage connections, enhance phylogenetic accuracy, and refine haplogroup classifications, ultimately providing a more complete picture of equine maternal evolution.

Beyond evolutionary and conservation implications, integrating haplotype and haplogroup classifications with known performance outcomes in different sports could reveal significant correlations between mtDNA haplogroups and performance traits. This could provide valuable insights for breeding and training programs, optimising selection strategies based on genetic predispositions. Notably, only two of the extracted articles attempted to explore this relationship [107,108].

Moving forward, advancing equine mtDNA research requires an integrated approach that combines whole mitochondrial genome sequencing, Y chromosome and autosomal markers, ancient DNA analysis, and standardised haplogroup classification. These efforts will be essential for improving phylogenetic resolution, enhancing conservation strategies, and uncovering the full extent of equine genetic diversity.

## 6. Conclusions

This systematic review provides a comprehensive synthesis of mtDNA research in horses, examining genetic diversity, maternal lineage tracing, and haplogroup classification across 76 studies. Our analysis identified major gaps and inconsistencies in the current literature, particularly in haplogroup classification, methodological approaches, and geographic representation.

Our findings reveal inconsistencies in mtDNA classification, with Achilli, Cieslak, and Jansen using different systems, leading to discrepancies in haplogroup assignments. Several haplotypes from Cieslak and Jansen remain unclassified in Achilli, highlighting gaps in the current frameworks. Additionally, trimming sequences for GenBank compatibility may cause data loss, while reliance on partial D-loop sequencing limits lineage resolution. Expanding analyses to include *CYTB*, *COX1*, or full mitochondrial genomes would improve classification accuracy. Geographical and breed biases further restrict our understanding of global equine diversity, with Sub-Saharan Africa, Oceania, and South America underrepresented. While well-documented breeds are frequently studied, many local and indigenous breeds lack genetic characterisation, limiting phylogenetic and conservation insights.

Future research should prioritise whole mitochondrial genome sequencing, the integration of Y chromosome and autosomal markers, and ancient DNA analysis to improve classification and trace extinct lineages. Expanding haplogroup reference databases will help standardise classification systems and resolve the existing gaps. Next-Generation Sequencing (NGS) provides a transformative solution, offering high accuracy, cost-effectiveness, and rapid sequencing of entire mitochondrial genomes. Its use in both modern and ancient DNA studies can refine phylogenetic relationships and uncover missing lineage connections. Finally, linking mtDNA haplogroups to performance traits remains an unexplored area, with only two studies investigating this connection. Further research could offer valuable insights for breeding and training strategies. Addressing these challenges will advance phylogenetics, conservation, and equine genetic research.

## Figures and Tables

**Figure 1 animals-15-00885-f001:**
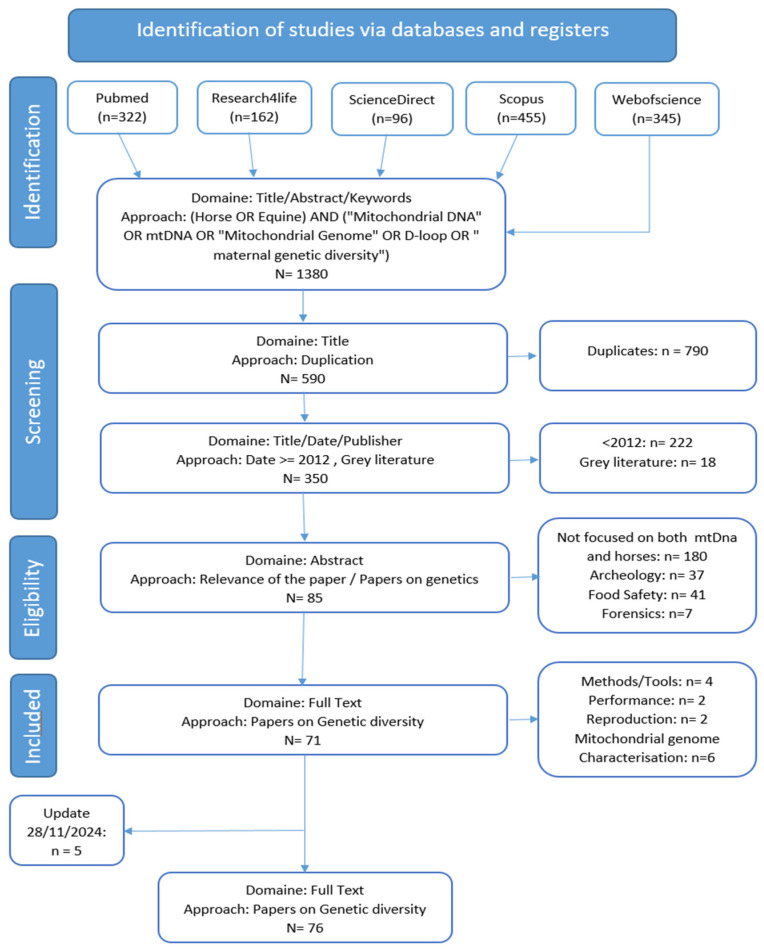
PRISMA flow diagram of study selection process.

**Figure 2 animals-15-00885-f002:**
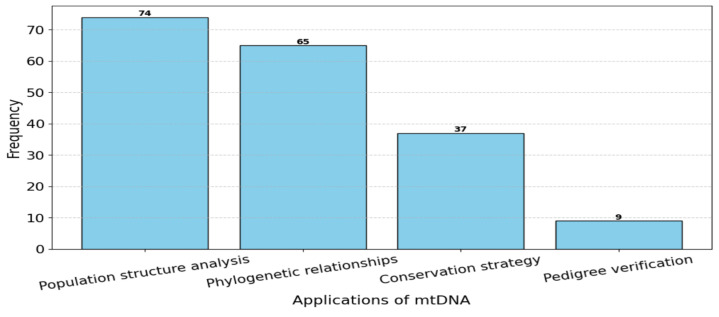
Histogram of mtDNA applications.

**Figure 3 animals-15-00885-f003:**
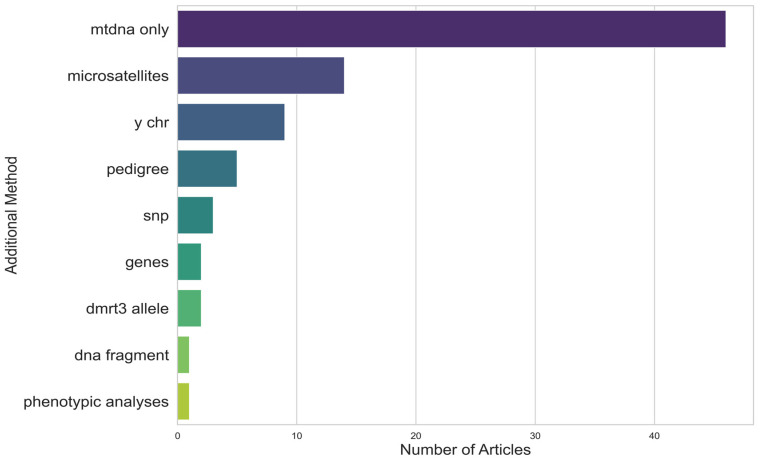
Frequency of additional genetic markers used in conjunction with mtDNA to study genetic diversity in horses.

**Figure 4 animals-15-00885-f004:**
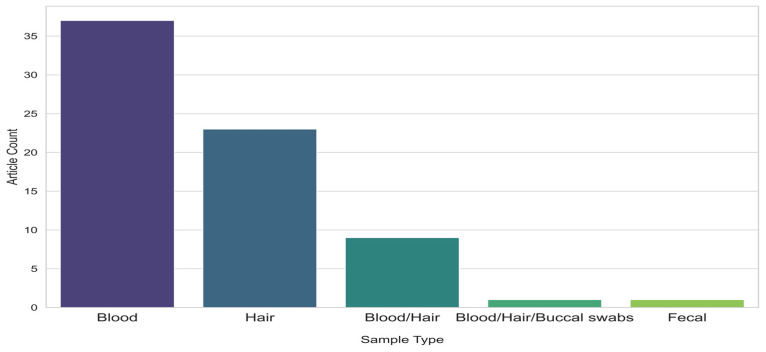
Distribution of the sample types used, according to the number of articles referring to them.

**Figure 5 animals-15-00885-f005:**
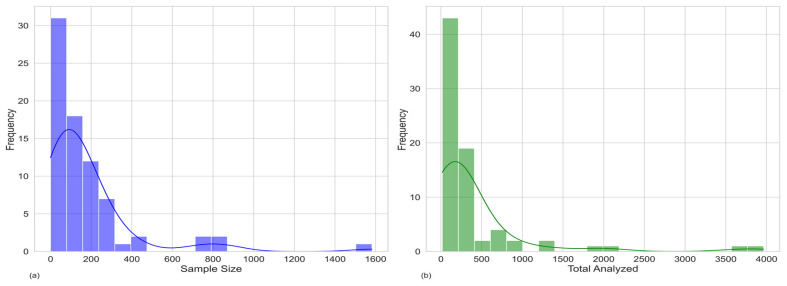
Histogram showing the frequency of different sample sizes. (**a**) The distribution without the addition of Genbank data; (**b**) the distribution of the total sample size.

**Figure 6 animals-15-00885-f006:**
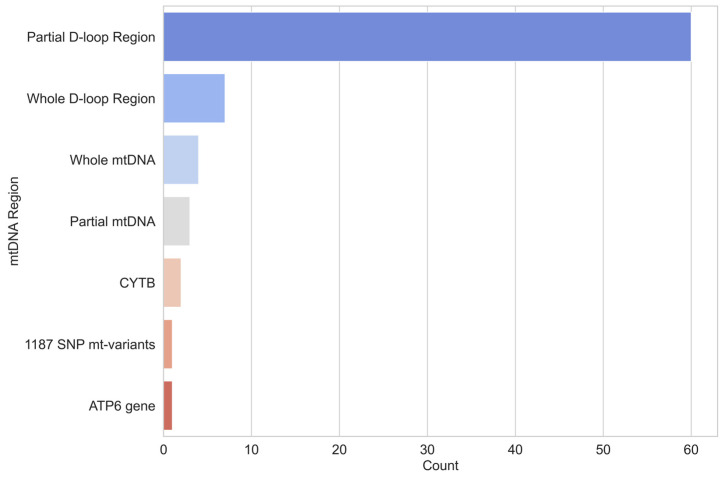
Distribution of mtDNA regions analysed.

**Figure 7 animals-15-00885-f007:**
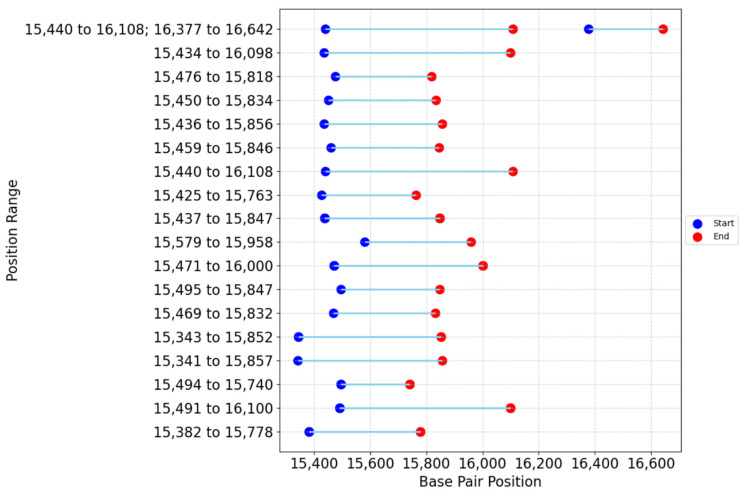
Position intervals of amplified portion in the partial D-loop region.

**Figure 8 animals-15-00885-f008:**
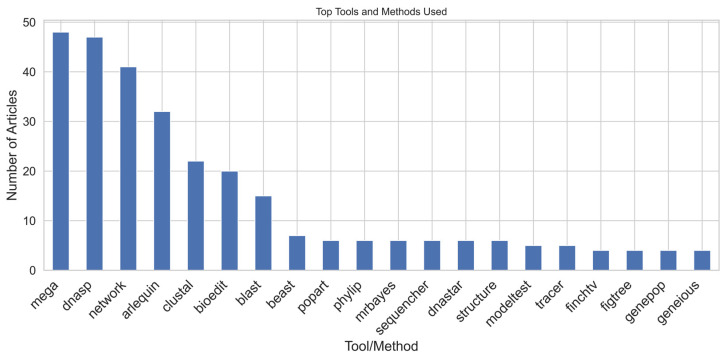
Frequency of tools and methods used in genetic diversity studies.

**Figure 9 animals-15-00885-f009:**
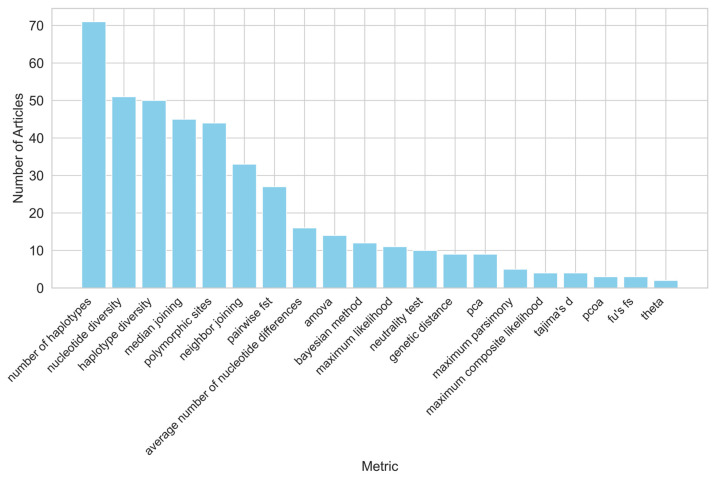
List of the metrics used and their frequency.

**Figure 10 animals-15-00885-f010:**
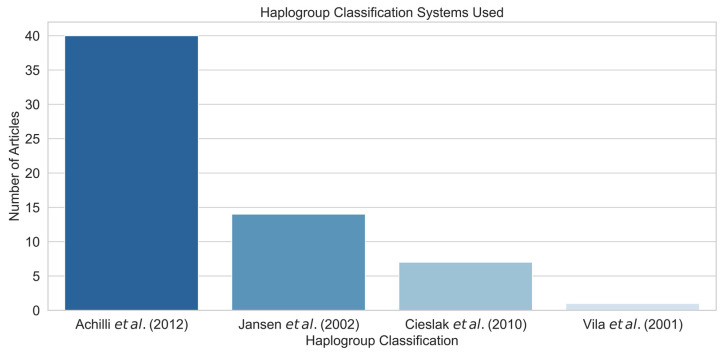
Haplogroup classification systems used [9,10,11,96].

**Table 1 animals-15-00885-t001:** Comparison of haplogroup classification systems for horse mtDNA.

Study	Base Pairs Sequenced	Number of Sequences	Number of Breeds	Haplogroups/Clusters
Vilà et al. (2001) **[96]**	355 bp	191 (+38 from GenBank)	10	6 clades (A to F)
Jansen et al. (2002) **[11]**	469 bp truncated to 247 bp	652 (318 new, 334 published)	25	17 clusters
Cieslak et al. (2010) **[10]**	722 bp truncated to 247 bp (HVR1)	1961 (207 ancient, 1754 modern)	46	19 haplogroups (A to K, X1 to X7)
Achilli et al. (2012) **[9]**	16,500 bp (whole mitochondrial genome)	83 complete mitochondrial genomes	Broad geographic representation	18 haplogroups (A to R)

**Table 2 animals-15-00885-t002:** Correspondence of haplogroup classification systems for horse mtDNA. In bold are the haplogroups that match in both phylogenetic trees (ME → Minimum Evolution; ML → Maximum Likelihood).

Achilli	Cieslak_ME	Jansen_ME	Cieslak_ML	Jansen_ML
A	**D2; D2b**		**D2; D2b;** G2	
B	**D; D3; D3a;**	**A3; A6**	**D; D3; D3a;**	**A3; A6**
C	**J**	**A6; B2**	**J**	**A6; B2**
D	**G; G3; Gx4**	**C2**	**G; G3; Gx4**	**C2**
E	**X3; X3d**		**X3; X3d**	
F		**A2**		**A2**
G				
H	**X4**; X6; X6c; **X12**; X14	**A6**	B3; **X4; X4a; X12**	**A6**
I	A; **I**	**B2**	**I**	**B2**
J	**H1; Kg2**		**H1; Kg2**	
K	**X5**	**A6**	**X5**	**A6**
L	**X1; X2; X2b**	**D1; D2; D3**	**X1; X2; X2b**	**D1; D2; D3**
M	**B**; X8a; **B1**	**C1**	**B; B1**	**C1**
N	**F**; G2	**C2**	**F**	**C2**
O	**K3**	**F1**	**K3**; K3b	**F1**
P	K3b	F1		
Q	**K2; K2b**	**F2**	**K2; K2b**	**F2**
R	**X7; X7a2; X7a; X7a1**	**G**	**X7; X7a2; X7a; X7a1**	**G**

## Data Availability

The original contributions presented in this study are included in the article and its Appendix A. Further inquiries can be directed to the corresponding author(s).

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
