# Peer review of "A Systematic Literature Review of Mitochondrial DNA Analysis for Horse Genetic Diversity"

_animals, 2025, doi:10.3390/ani15060885_

Round 1

Reviewer 1 Report

Comments and Suggestions for Authors

Comments for Authors

The manuscript, titled "A Systematic Literature Review of Mitochondrial DNA Analysis for Horse Genetic Diversity," presents a commendable effort to synthesize current research on equine mtDNA analysis, focusing on genetic variation, maternal lineage tracing, and haplogroup identification. The use of the PRISMA methodology, inclusion of 76 studies from 2012 onwards (updated to November 2024), and identification of key gaps—such as inconsistent haplogroup classifications and underrepresented breeds—are notable strengths. The review addresses a relevant topic in equine population genetics and conservation, offering a solid foundation for future research directions. However, several significant issues related to structure, depth of analysis, clarity, and presentation prevent its acceptance in its current form. I recommend Major Revision to address these concerns, as outlined below, to enhance the manuscript’s scientific rigor and readability for publication.

Major Comments

The "Results and Discussion" section combines empirical findings (e.g., sample types, sequencing methods) with their interpretation, which muddles the distinction between data presentation and analysis. Most journals expect these sections to be separate to ensure clarity and adherence to scientific reporting standards.

Recommendation: Split this into two distinct sections: "Results" for objective data (e.g., "53% of studies used blood samples," "no correlations found") and "Discussion" for interpretation and literature comparison (e.g., implications of trimming, significance of Sanger dominance). This will improve readability and align with academic norms.

Several findings lack sufficient discussion or context. For example: The absence of meaningful correlations in the correlation matrix (sample size, GenBank sequences, etc.) is stated but not explored—why might this be the case (e.g., study heterogeneity, methodological differences)?

The variety of haplogroup classification systems is highlighted, but their practical implications (e.g., on phylogenetic accuracy or conservation planning) are underexplored.

Deepen the interpretation by discussing potential reasons for key findings and their broader significance. For instance, speculate on why correlations were absent (e.g., diverse study designs) and elaborate on how classification inconsistencies affect cross-study synthesis or equine genetic research.

While the English is generally sound, the academic tone is inconsistent, and some sections suffer from verbosity or repetition. Examples include:

"Can result in the loss of valuable information" (Conclusions) could be more assertive ("results in loss").

The potential of next-generation sequencing (NGS) is mentioned multiple times without adding new insight.

Long sentences (e.g., in Sampling) reduce readability.

Revise for conciseness and a stronger academic tone. Shorten sentences, eliminate redundancies (e.g., consolidate NGS mentions), and use definitive language where appropriate (e.g., "is" instead of "can be"). A thorough proofreading is advised.

Some methodological aspects lack specificity, reducing reproducibility. For example:

The "standardized form" for data extraction is mentioned, but its structure or fields are not described.

The update process (November 2024, adding 5 articles) lacks detail on how these studies were identified and screened.

Provide more detail on the data extraction form (e.g., list key variables included) and clarify the update process (e.g., same keyword search reapplied?). This will strengthen the methodological transparency.

The Conclusions section effectively summarizes challenges and future directions but lacks specificity and a strong closing impact. For instance, "more studies utilizing other parts of the mtDNA" is vague, and the link to practical outcomes (e.g., conservation) could be more compelling.

Specify which mtDNA regions (e.g., CYTB, full D-loop) warrant focus, and elaborate on underrepresented regions/breeds (e.g., name examples like Sub-Saharan Africa). End with a clear call-to-action tying findings to conservation or breeding applications, emphasizing the review’s real-world relevance.

Minor Comments

- Use "mtDNA" consistently after its first full mention (e.g., avoid switching to "mitochondrial DNA").

- Standardize percentages (e.g., "53%" not "53 %") and numbers (e.g., "1,380" not "1380").

- Some claims (e.g., non-invasive methods’ superiority [16]) are broad; provide specific examples or data from cited studies to bolster credibility.

- Ensure all references align with claims (e.g., check [47,48] fully support performance trait correlations).

- The Introduction mentions 2012 as a cutoff but doesn’t justify it (e.g., technological advances?). Add a brief rationale.

- The Sampling section notes wide ranges (2–1,582 without GenBank, 17–3,965 with), but the implications (e.g., statistical power) are not discussed. Briefly address this.

Comments on the Quality of English Language

The English language quality in the manuscript "A Systematic Literature Review of Mitochondrial DNA Analysis for Horse Genetic Diversity" is generally average, with a clear intent and structure that allows the scientific content to be understood. However, there are several areas where improvements are needed to enhance readability, strengthen the academic tone, and meet the expectations of a high-quality journal publication.

Author Response

Reviewer 1

  • All changes made in the manuscript were highlighted in yellow.

Major Comments

Comment 1/ The "Results and Discussion" section combines empirical findings (e.g., sample types, sequencing methods) with their interpretation, which muddles the distinction between data presentation and analysis. Most journals expect these sections to be separate to ensure clarity and adherence to scientific reporting standards.

Recommendation: Split this into two distinct sections: "Results" for objective data (e.g., "53% of studies used blood samples," "no correlations found") and "Discussion" for interpretation and literature comparison (e.g., implications of trimming, significance of Sanger dominance). This will improve readability and align with academic norms.

Response: 

  • We split the results and discussion.

Comment 2/ Several findings lack sufficient discussion or context. For example: The absence of meaningful correlations in the correlation matrix (sample size, GenBank sequences, etc.) is stated but not explored—why might this be the case (e.g., study heterogeneity, methodological differences)?

The variety of haplogroup classification systems is highlighted, but their practical implications (e.g., on phylogenetic accuracy or conservation planning) are underexplored.

Deepen the interpretation by discussing potential reasons for key findings and their broader significance. For instance, speculate on why correlations were absent (e.g., diverse study designs) and elaborate on how classification inconsistencies affect cross-study synthesis or equine genetic research.

Response: 

  • For correlation, we added a paragraph in the Discussion section (lines 432-440) to explore potential reasons behind the absence of significant correlations.

Added text:

"Most of the articles followed a clear methodological pipeline, yet significant differences were observed in the specific approaches used at each step. No correlation was found between the key variables extracted, suggesting that methodological choices were shaped by diverse factors. For instance, sample sizes are influenced by the size of the existing population, the access to the horses or their biological material, and the specific objectives of each study—whether aiming for broad comparative analyses or focusing on a single population. Similarly, the number of base pairs sequenced varied, potentially influenced by financial constraints, researcher preferences, or the need to align with available reference sequences in GenBank."

This addition provides context on why correlations were absent, linking it to study heterogeneity, sample availability, and methodological differences.

  • For Haplogroups: we conducted additional analyses, including phylogenetic tree construction and a correspondence table, to compare haplogroups across classification systems. The following changes were made:

Methodology section (lines 128-136):

Describes how we compared mitochondrial haplogroup classifications by aligning sequences, selecting representative sequences, trimming them to 386 bp, and constructing phylogenetic trees using MEGA v12.

"To compare mitochondrial haplogroup classifications (Achilli, Cieslak, and Jansen), we aligned representative sequences from each system. Some Jansen haplogroups were unavailable, so we selected accessible sequences and trimmed them to 386 base pairs for consistency. Phylogenetic trees were constructed with MEGA version 12 [13] using Minimum Evolution (500 bootstrap iterations) and Maximum Likelihood (Tamura-Nei model, 500 bootstrap iterations). The Tamura-Nei was provided by the program implemented in MEGA vers 12. These analyses produced a correspondence table linking haplogroups across classifications."

Results section (lines 380-400):

Details the construction of phylogenetic trees and the creation of a correspondence table.

"We generated phylogenetic trees for the three classification systems (Achilli, Cieslak, and Jansen), using Achilli’s system as the reference (Figures S1, S2). These trees allowed us to establish a correspondence table (Table 2) aligning haplogroups from Cieslak and Jansen with those of Achilli. For most haplogroups, the correspondence was consistent between the two phylogenetic methods used—minimum evolution (ME) and maximum likelihood (ML). However, some discrepancies were observed; for instance, Cieslak’s K3b corresponded to haplogroup O in ML but to P in ME. Several haplogroups from Jansen and Cieslak had no equivalent in Achilli’s classification. Notably, A1, identified in Jansen, lacked an assigned cluster in Achilli’s system. Cieslak’s classification also included extinct haplogroups, such as X3c1a, X6c, X8a, X13, X14, G2, and B3. Some of these haplogroups (G2, B3, X14, X6c, and X8a) had correspondences in one or both phylogenetic techniques, while others (X3c1a and X13) had no equivalent in Achilli’s classification. Additionally, certain haplogroups still present in modern horses, such as A, K, K1, and X3c1, remained unclassified in Achilli’s system in at least one method (K1 and A in one, and K and X3c1 in both). Moreover, our review highlighted that several studies have reported haplotypes that do not fit into any of the existing classification systems [49–52]."

A correspondence table (Table 2) was added to align haplogroups from different systems.

Discussion section (lines 518-528):

“Our results showed that several non-extinct haplogroups from Cieslak and Jansen had no equivalent in Achilli’s classification, and some studies identified haplotypes that do not fit into any known system. This suggests that Achilli’s framework may be incomplete and that our broader understanding of equine mitochondrial lineages re-mains limited. The variety of haplogroup classification systems reflects ongoing efforts to characterize equine maternal lineages, yet their inconsistencies pose challenges for comparative studies. Differences in nomenclature, resolution, and reference datasets can impact phylogenetic accuracy, lineage tracing, and conservation planning. The ab-sence of standardized criteria may lead to discrepancies in haplogroup assignment, limiting the reproducibility of findings across studies. Furthermore, the incompleteness of existing systems, as seen in our results, suggests that some mitochondrial lineages remain unclassified, which may lead to their loss from genetic records.“

  • We also added a limitation and perspective part. (541-616) (more details in comment n°5)

Comment 3/ While the English is generally sound, the academic tone is inconsistent, and some sections suffer from verbosity or repetition. Examples include:

"Can result in the loss of valuable information" (Conclusions) could be more assertive ("results in loss").

The potential of next-generation sequencing (NGS) is mentioned multiple times without adding new insight.

Long sentences (e.g., in Sampling) reduce readability.

Revise for conciseness and a stronger academic tone. Shorten sentences, eliminate redundancies (e.g., consolidate NGS mentions), and use definitive language where appropriate (e.g., "is" instead of "can be"). A thorough proofreading is advised.

Response: 

  • We carefully revised the manuscript to shorten long sentences, use more assertive language and ensure a consistent academic tone throughout.

Comment 4/ Some methodological aspects lack specificity, reducing reproducibility. For example:

The "standardized form" for data extraction is mentioned, but its structure or fields are not described.

The update process (November 2024, adding 5 articles) lacks detail on how these studies were identified and screened.

Response: 

  • To clarify this, we changed lines 103-104 in methodology. The standardized form, is us extracting the same variables and data from all the articles and organizing them into a table as mentioned in lines 108-124.
  • We added more details about the update process. Lines 95-98.

Comment 5/ Provide more detail on the data extraction form (e.g., list key variables included) and clarify the update process (e.g., same keyword search reapplied?). This will strengthen the methodological transparency.

The Conclusions section effectively summarizes challenges and future directions but lacks specificity and a strong closing impact. For instance, "more studies utilizing other parts of the mtDNA" is vague, and the link to practical outcomes (e.g., conservation) could be more compelling.

Specify which mtDNA regions (e.g., CYTB, full D-loop) warrant focus, and elaborate on underrepresented regions/breeds (e.g., name examples like Sub-Saharan Africa). End with a clear call-to-action tying findings to conservation or breeding applications, emphasizing the review’s real-world relevance.

Response: 

  • We added a section called Limitations and Perspective to give more details (lines 541-616).

Limitations:

“Several limitations must be considered in interpreting our findings. First, in our comparison of haplogroup classification systems, sequences were trimmed to 386 base pairs to ensure consistency across datasets. This reduction may have led to information loss, potentially affecting haplogroup correspondence. Many studies also trimmed their sequences to align with GenBank references, a practice that can obscure genetic variation.

Another limitation is the reliance on mtDNA alone for genetic characterization. While mtDNA is widely used for maternal lineage tracing, it does not capture paternal ancestry or autosomal variation, limiting the broader understanding of genetic diversity. Even within mtDNA-based studies, most rely on partial D-loop sequences, despite the availability of other informative mitochondrial markers such as cytochrome b (CYTB) and protein-coding genes. This narrow focus reduces the accuracy of haplogroup assignments and overlooks phylogenetic relationships that could be better resolved with whole mitochondrial genomes.

Geographical and breed representation in equine mtDNA studies remains skewed, limiting the global understanding of mitochondrial diversity. Certain regions remain underrepresented, despite their potential to reveal unique and uncharacterized maternal lineages. Similarly, research has largely focused on a few well-known breeds, while many local and indigenous breeds remain unexamined. Addressing these gaps is crucial for obtaining a complete picture of equine maternal diversity.

The variety of haplogroup classification systems further complicates comparative studies. Several haplotypes identified in Cieslak and Jansen remain unclassified in Achilli, suggesting that current classification frameworks are incomplete. Additionally, some studies reported haplotypes that do not fit into any established system, emphasizing the need for a unified and expanded classification approach to improve consistency and reproducibility in equine mtDNA research.”

Perspectives:

“Future studies should prioritize the use of additional genetic markers and longer mtDNA sequences to enhance the resolution of lineage characterization. Expanding beyond partial D-loop sequencing to include CYTB, COX1 gene, and potentially full mitochondrial genomes would improve haplogroup assignment and phylogenetic accuracy. Revisiting previously studied breeds using longer sequences could also minimize data loss from sequence trimming and improve genetic profiling.

Expanding research efforts to underrepresented breeds and regions is essential for a more comprehensive understanding of equine mitochondrial diversity. Increasing sampling from Africa, South America, and Asia could uncover previously undocumented haplotypes and refine existing classifications.

Ancient DNA (aDNA) research provides another critical avenue for mitochondrial lineage studies. Some extinct lineages identified in Cieslak and Jansen remain unclassified in Achilli, raising questions about how ancient sequences integrate into modern haplogroup frameworks. Future studies should compare ancient D-loop sequences from Cieslak and Jansen against Achilli’s dataset to determine which lineages can be classified and which remain unresolved. Given the limitations of short sequences for such comparisons, whole mitochondrial genome sequencing of ancient remains should be prioritized to provide a more robust dataset for phylogenetic analysis.

A major step toward improving mitochondrial studies is the standardization and unification of haplogroup nomenclature. The discrepancies between Achilli, Cieslak, and Jansen highlight the need for an updated and expanded classification system that integrates both modern and ancient mitochondrial sequences. Expanding haplogroup reference databases by sequencing additional complete mitochondrial genomes, particularly from breeds reporting unclassified haplotypes, would refine classification accuracy and bridge gaps between existing systems.

Finally, Next-Generation Sequencing (NGS) presents a powerful tool for addressing these research gaps. NGS generates significantly more data than Sanger sequencing in shorter timeframes, making it highly efficient for whole mitochondrial genome analysis. Its accuracy and cost-effectiveness allow for the simultaneous sequencing of multiple mtDNA genomes, providing deeper insights into genetic diversity and phylogenetic relationships. NGS is particularly valuable for degraded and ancient samples, as demonstrated in paleogenomic research [64]. Applying NGS to ancient horse remains could clarify missing lineage connections, enhance phylogenetic accuracy, and refine haplogroup classifications, ultimately providing a more complete picture of equine maternal evolution.

Beyond evolutionary and conservation implications, integrating haplotype and haplogroup classifications with known performance outcomes in different sports could reveal significant correlations between mtDNA haplogroups and performance traits. This could provide valuable insights for breeding and training programs, optimizing selection strategies based on genetic predispositions. Notably, only two of the extracted articles attempted to explore this relationship [65,66].

Moving forward, advancing equine mtDNA research requires an integrated approach that combines whole mitochondrial genome sequencing, Y-chromosome and autosomal markers, ancient DNA analysis, and standardized haplogroup classification. These efforts will be essential for improving phylogenetic resolution, enhancing conservation strategies, and uncovering the full extent of equine genetic diversity.”

  • We also modified the conclusion to synthetize what we found in this review (lines 621-648).

“This systematic review provides a comprehensive synthesis of mtDNA research in horses, examining genetic diversity, maternal lineage tracing, and haplogroup classification across 76 studies. Our analysis identified major gaps and inconsistencies in the current literature, particularly in haplogroup classification, methodological approaches, and geographic representation.

Our findings reveal inconsistencies in mtDNA classification, with Achilli, Cieslak, and Jansen using different systems, leading to discrepancies in haplogroup assignments. Several haplotypes from Cieslak and Jansen remain unclassified in Achilli, highlighting gaps in current frameworks. Additionally, trimming sequences for GenBank compatibility may cause data loss, while reliance on partial D-loop sequencing limits lineage resolution. Expanding analyses to include CYTB, COX1, or full mitochondrial genomes would improve classification accuracy. Geographical and breed biases further restrict our understanding of global equine diversity, with Sub-Saharan Africa, Oceania, and South America underrepresented. While well-documented breeds are frequently studied, many local and indigenous breeds lack genetic characterization, limiting phylogenetic and conservation insights.

Future research should prioritize whole mitochondrial genome sequencing, the integration of Y-chromosome and autosomal markers, and ancient DNA analysis to improve classification and trace extinct lineages. Expanding haplogroup reference databases will help standardize classification systems and resolve existing gaps. Next-Generation Sequencing (NGS) provides a transformative solution, offering high accuracy, cost-effectiveness, and rapid sequencing of entire mitochondrial genomes. Its use in both modern and ancient DNA studies can refine phylogenetic relationships and uncover missing lineage connections. Finally, linking mtDNA haplogroups to performance traits remains an unexplored area, with only two studies investigating this connection. Further research could offer valuable insights for breeding and training strategies. Addressing these challenges will advance phylogenetics, conservation, and equine genetic research.”

Minor Comments

Comment 6/ Use "mtDNA" consistently after its first full mention (e.g., avoid switching to "mitochondrial DNA").

Response: 

  • Done

Comment 7/ Standardize percentages (e.g., "53%" not "53 %") and numbers (e.g., "1,380" not "1380").

Response: 

  • Done

Comment 8/ Some claims (e.g., non-invasive methods’ superiority [16]) are broad; provide specific examples or data from cited studies to bolster credibility.

Response: 

  • We included some examples and more details in Lines (444 – 464).

Blood was the most commonly used sample because it provides sufficient high-quality DNA for genetic analysis, though collection can sometimes be challenging. Non-invasive sampling methods —including faeces, hair, saliva—are easier to collect and less invasive than blood. As a result, they are increasingly used to address animal welfare concerns [55]. In a systematic literature review, Zemanova et al.[55] reported a steady increase in the use of non-invasive methods in wildlife research. Comparative studies have evaluated DNA isolation efficiency across different sample types. Gurău et al. [56] compared DNA extraction from blood and hair follicle samples in goats, finding that while whole blood yielded slightly higher DNA quantities, both sample types provided sufficient DNA for amplifying 1,200 bp fragments. Similarly, Muko et al. [57] assessed DNA yield and quality from oral mucosa swabs and faeces in horses, comparing them with DNA from blood. Although non-invasive samples provided sufficient DNA for PCR analysis, contamination from microorganisms and feed remained a concern.

In a review of 113 studies, Zemanova et al. [55] found that 94% of studies reported equivalent or superior performance of non-invasive genetic assessments compared to invasive sampling. Non-invasive techniques can also be more cost-effective and time-efficient. Thus, adopting non-invasive extraction methods could simplify DNA collection while maintaining comparable DNA quality. This approach may also achieve similar sequencing performance for the entire D-loop region of mtDNA, which is approximately 1,192 bp long [17]. “

Comment 9/ Ensure all references align with claims (e.g., check [47,48] fully support performance trait correlations).

Response: 

  • [47] -> [65] in the latest version is the article by Engel et al., in their abstract they said: “Certain mitochondrial haplogroups were associated with special talents for dressage or show jumping. “
    [48] -> [66] by Lin et al. said : “The results suggested that haplogroup L3b, could have a negative association with elite performance. The T1458C mutation harboured in haplogroup L3b could have a functional effect that is related to poor athletic performance.”

Comment 10/ The Introduction mentions 2012 as a cutoff but doesn’t justify it (e.g., technological advances?). Add a brief rationale.

Response: 

  • We added this to the methodology (lines 78-80):

“This cut-off was chosen because 2012 marks the publication of Achilli et al.'s [9] classi-fication system, which remains the most recent and comprehensive framework for eq-uine mtDNA haplogroup classification.”

Comment 11/ The Sampling section notes wide ranges (2–1,582 without GenBank, 17–3,965 with), but the implications (e.g., statistical power) are not discussed. Briefly address this.

Response: 

  • We added this to the discussion (465-471):

“The sample sizes varied widely, with some studies focusing on nearly extinct breeds with fewer than 20 individuals, while others aimed to assess the global distribution of horses, incorporating thousands of sequences into their analyses. Smaller sample sizes may limit the ability to detect rare haplotypes, potentially underestimating genetic diversity within a population. Conversely, larger sample sizes generally enhance the statistical power of analyses but may introduce biases if certain breeds or regions are disproportionately represented.”

Reviewer 2 Report

Comments and Suggestions for Authors

Review on the manuscript titled “A Systematic literature review of mitochondrial DNA analysis for Horse Genetic Diversity” by Agbani et al., 2025.

                The authors target the area of Horse Genetic Diversity/Ancestry based on mtDNA analysis.

In the course of the review, the authors outlined multiple stats.

                The protocol of materials retrieval is described in Materials section, and comprised 1380 by the initial query: ("Mitochondrial DNA" OR "mtDNA" OR "Mitochondrial Genome" OR "D-loop" OR "Maternal genetic diversity") on July 2023 followed update in 2024 for 5 articles. It followed the filtering/categorizing process depicted in Flow Diagram of Study Selection/Filtering  Process (Fig. 1), which led to 76 non-redundant relevant publications.  Haplogroups were deposited in table S1.

                Next followed the stat on the various features of mtDNA traits and data types:

  • “Figure 2. Frequency of additional genetic markers used in conjunction with mtDNA to study genetic diversity in horses. (Various ancestry genetic markers other than mtDNA)
  • Figure 3. “Distribution of the sample types used, according to the number of articles referring to them” depicts major blood samples and some other types.
  • Figure 4. Histogram showing the frequency of different sample sizes.
  • 2/3.3. DNA extraction/PCR Amplification features discussion and region amplification type (Fig. 5; the partial D-loop region is the major target) with Fig.6 depicting major ‘Position intervals of amplified portion in the partial D-loop region’.
  • “3.4. Sequencing techniques” reveals that major sequencing technique is Sanger sequencing (3130 Genetic Analyzer and the 3730xl 96-capillary DNA Analyzer), followed with some NGS sequencing instances.
  • Analytical tools chapter: 20 analytical software suites used are presented in Fig. 7
  • 6. Metrics. : It refers to the target heterogeneity values which was commonly assessed by # haplotypes (93% of instances) and some others (Fig. 8).
  • 7. Haplogroups: Haplogroups classification system was majorly used from Achilli et al., 2012 (40% of instances) and some others (Fig. 9; Table 1: Comparison of Haplogroup Classification Systems for Horse mtDNA.).
  • 8. Global Distribution of studies: The authors Provided the distribution of studies presented in Figure S1.
  • 9.’ Breeds Studied’ chapter revealed the Arabian Horse is most studied species (Table S2).

As a conclusion, the authors report: “Finally, integrating haplotype and haplogroup classifications with known performance outcomes in different sports could reveal significant correlations between mtDNA haplogroups and performance traits, providing valuable insights for breeding and training programs.”

Overall, the manuscript may be of interest to the researchers/breeders in the field. Some notes are listed below.

There are problems with supplements:

  • 1 is outlined as a movie which is not comprehensive/adequate given the speed rate. Please make it either in a still format (best), or reduce the motion speed rate significantly.
  • Based on the material collected, it would be extremely relevant to present a haplotypes tree.
  • A table of haplogroups observed should be presented in the manuscript/supplement.
  • Total genetics.rdf should be reformatted in pdf format as a much more common one; I was unable immediately convert it.
  • Table S1 is in the protected format and is blocked for viewing – please amend all limitations on viewing.
  • Both supplementary Figures should be made more contrasted in color for a clear view.

Author Response

Reviewer 2

  • All changes made in the manuscript were highlighted in yellow.

Comment 1/   1 is outlined as a movie which is not comprehensive/adequate given the speed rate. Please make it either in a still format (best), or reduce the motion speed rate significantly.

Response:

  • We changed it to be still, the reader can manually slide to see each Haplogroup or click on the Play button for it to be in motion (we slowed it to about 5s per Haplogroup).

Comment 2/ Based on the material collected, it would be extremely relevant to present a haplotypes tree.

Response:

  • We conducted additional analyses, including phylogenetic tree construction and a correspondence table, to compare haplogroups across classification systems. The following changes were made:

Methodology section (lines 128-136):

Describes how we compared mitochondrial haplogroup classifications by aligning sequences, selecting representative sequences, trimming them to 386 bp, and constructing phylogenetic trees using MEGA v12.

"To compare mitochondrial haplogroup classifications (Achilli, Cieslak, and Jansen), we aligned representative sequences from each system. Some Jansen haplogroups were unavailable, so we selected accessible sequences and trimmed them to 386 base pairs for consistency. Phylogenetic trees were constructed with MEGA version 12 [13] using Minimum Evolution (500 bootstrap iterations) and Maximum Likelihood (Tamura-Nei model, 500 bootstrap iterations). The Tamura-Nei was provided by the program implemented in MEGA vers 12. These analyses produced a correspondence table linking haplogroups across classifications."

Results section (lines 380-400):

Details the construction of phylogenetic trees and the creation of a correspondence table.

"We generated phylogenetic trees for the three classification systems (Achilli, Cieslak, and Jansen), using Achilli’s system as the reference (Figures S1, S2). These trees allowed us to establish a correspondence table (Table 2) aligning haplogroups from Cieslak and Jansen with those of Achilli. For most haplogroups, the correspondence was consistent between the two phylogenetic methods used—minimum evolution (ME) and maximum likelihood (ML). However, some discrepancies were observed; for instance, Cieslak’s K3b corresponded to haplogroup O in ML but to P in ME. Several haplogroups from Jansen and Cieslak had no equivalent in Achilli’s classification. Notably, A1, identified in Jansen, lacked an assigned cluster in Achilli’s system. Cieslak’s classification also included extinct haplogroups, such as X3c1a, X6c, X8a, X13, X14, G2, and B3. Some of these haplogroups (G2, B3, X14, X6c, and X8a) had correspondences in one or both phylogenetic techniques, while others (X3c1a and X13) had no equivalent in Achilli’s classification. Additionally, certain haplogroups still present in modern horses, such as A, K, K1, and X3c1, remained unclassified in Achilli’s system in at least one method (K1 and A in one, and K and X3c1 in both). Moreover, our review highlighted that several studies have reported haplotypes that do not fit into any of the existing classification systems [49–52]."

A correspondence table (Table 2) was added to align haplogroups from different systems.

Discussion section (lines 518-528):

Our results showed that several non-extinct haplogroups from Cieslak and Jansen had no equivalent in Achilli’s classification, and some studies identified haplotypes that do not fit into any known system. This suggests that Achilli’s framework may be incomplete and that our broader understanding of equine mitochondrial lineages re-mains limited. The variety of haplogroup classification systems reflects ongoing efforts to characterize equine maternal lineages, yet their inconsistencies pose challenges for comparative studies. Differences in nomenclature, resolution, and reference datasets can impact phylogenetic accuracy, lineage tracing, and conservation planning. The ab-sence of standardized criteria may lead to discrepancies in haplogroup assignment, limiting the reproducibility of findings across studies. Furthermore, the incompleteness of existing systems, as seen in our results, suggests that some mitochondrial lineages remain unclassified, which may lead to their loss from genetic records.

  • We also added a limitation and perspective part. (541-616). The trees are in supplementary data as Figure S1 and Figure S2.

Comment 3/ A table of haplogroups observed should be presented in the manuscript/supplement.

Response:

  • A table listing the observed haplogroups was already included in Table S1, where we documented haplogroups identified in each study. Additionally, we have now added a correspondence table that aligns haplogroups across different classification systems (Achilli, Cieslak, and Jansen)

Comment 4/ Total genetics.rdf should be reformatted in pdf format as a much more common one; I was unable immediately convert it.

Response:

  • It’s the file to get the references directly to zotero. We have also exported the bibliography as a Html and added it to the supplementary data.

Comment 5/ Table S1 is in the protected format and is blocked for viewing – please amend all limitations on viewing.

Response:

  • In the status of the table it says ‘Unprotected’, I don’t know why the reviewer can’t open it. To ensure easy access, we have also provided Table S1 as a direct download link: https://docs.google.com/spreadsheets/d/1XVLF45uSLDtx8KxfofjbFwuHlzugVyBI/edit?usp=drive_link&ouid=102396531136433746721&rtpof=true&sd=true.

Please let us know if any issues persist.

Comment 6/ Both supplementary Figures should be made more contrasted in color for a clear view.

Response:

  • We changed the colors for both figures to get a clearer view.

Reviewer 3 Report

Comments and Suggestions for Authors

Mitochondrial DNA analysis serves as an essential tool for assessing genetic diversity and the evolutionary history of horses. The authors have presented a comprehensive review on this topic. However, some revisions are required to incorporate recent advancements and improve the overall quality of the manuscript.
1. The authors have provided a broad insight into how they collected the articles, organized them based on methodology, gene ontology, and the obtained results of different studies. Most reviewers follow a similar approach, but as researchers, it is important to highlight the unique findings and conclusions drawn from the review. This critical aspect is missing in the manuscript.
2. The authors focused only on mtDNA D-loop variation, without discussing the Cytochrome b gene, COX1 gene, and other coding regions of mtDNA. In my opinion, these regions should also be included in the review to provide a comprehensive overview of the manuscript.
3. In my opinion, instead of discussing the review collection methodology, the authors should focus on how the review would contribute to future research, especially in the areas of phylogenetic differentiation or sequence variations among horse breeds such as Arabian, Thoroughbred, and Mongolian. The review should also highlight its application in domestication events, migration patterns, and breed evolution. Although mtDNA is primarily used to study the maternal lineages of different horse breeds, the authors have missed providing information on predominant research works related to population structure analysis and the phylogenetic evolutionary relationship between horse breeds.
4. The review may be modified to focus on the current challenges, applications of mtDNA research, future research perspectives, and limitations of mtDNA use in research work.

Lines 190-194: The review by Gurău et al. [15] discusses non-invasive DNA isolation from goat samples, which is not directly relevant to the manuscript. It would be more appropriate to replace this with studies focusing on non-invasive DNA isolation techniques in horses, such as hair follicles or buccal swabs, to better align with the manuscript's scope.

Author Response

Reviewer 3

  • All changes made in the manuscript were highlighted in yellow.

Comment 1/ The authors have provided a broad insight into how they collected the articles, organized them based on methodology, gene ontology, and the obtained results of different studies. Most reviewers follow a similar approach, but as researchers, it is important to highlight the unique findings and conclusions drawn from the review. This critical aspect is missing in the manuscript.

Response: 

  • We added a section called Limitations and Perspective to give more details (lines 541-616).

Limitations:

“Several limitations must be considered in interpreting our findings. First, in our comparison of haplogroup classification systems, sequences were trimmed to 386 base pairs to ensure consistency across datasets. This reduction may have led to information loss, potentially affecting haplogroup correspondence. Many studies also trimmed their sequences to align with GenBank references, a practice that can obscure genetic variation.

Another limitation is the reliance on mtDNA alone for genetic characterization. While mtDNA is widely used for maternal lineage tracing, it does not capture paternal ancestry or autosomal variation, limiting the broader understanding of genetic diversity. Even within mtDNA-based studies, most rely on partial D-loop sequences, despite the availability of other informative mitochondrial markers such as cytochrome b (CYTB) and protein-coding genes. This narrow focus reduces the accuracy of haplogroup assignments and overlooks phylogenetic relationships that could be better resolved with whole mitochondrial genomes.

Geographical and breed representation in equine mtDNA studies remains skewed, limiting the global understanding of mitochondrial diversity. Certain regions remain underrepresented, despite their potential to reveal unique and uncharacterized maternal lineages. Similarly, research has largely focused on a few well-known breeds, while many local and indigenous breeds remain unexamined. Addressing these gaps is crucial for obtaining a complete picture of equine maternal diversity.

The variety of haplogroup classification systems further complicates comparative studies. Several haplotypes identified in Cieslak and Jansen remain unclassified in Achilli, suggesting that current classification frameworks are incomplete. Additionally, some studies reported haplotypes that do not fit into any established system, emphasizing the need for a unified and expanded classification approach to improve consistency and reproducibility in equine mtDNA research.”

Perspectives:

“Future studies should prioritize the use of additional genetic markers and longer mtDNA sequences to enhance the resolution of lineage characterization. Expanding beyond partial D-loop sequencing to include CYTB, COX1 gene, and potentially full mitochondrial genomes would improve haplogroup assignment and phylogenetic accuracy. Revisiting previously studied breeds using longer sequences could also minimize data loss from sequence trimming and improve genetic profiling.

Expanding research efforts to underrepresented breeds and regions is essential for a more comprehensive understanding of equine mitochondrial diversity. Increasing sampling from Africa, South America, and Asia could uncover previously undocumented haplotypes and refine existing classifications.

Ancient DNA (aDNA) research provides another critical avenue for mitochondrial lineage studies. Some extinct lineages identified in Cieslak and Jansen remain unclassified in Achilli, raising questions about how ancient sequences integrate into modern haplogroup frameworks. Future studies should compare ancient D-loop sequences from Cieslak and Jansen against Achilli’s dataset to determine which lineages can be classified and which remain unresolved. Given the limitations of short sequences for such comparisons, whole mitochondrial genome sequencing of ancient remains should be prioritized to provide a more robust dataset for phylogenetic analysis.

A major step toward improving mitochondrial studies is the standardization and unification of haplogroup nomenclature. The discrepancies between Achilli, Cieslak, and Jansen highlight the need for an updated and expanded classification system that integrates both modern and ancient mitochondrial sequences. Expanding haplogroup reference databases by sequencing additional complete mitochondrial genomes, particularly from breeds reporting unclassified haplotypes, would refine classification accuracy and bridge gaps between existing systems.

Finally, Next-Generation Sequencing (NGS) presents a powerful tool for addressing these research gaps. NGS generates significantly more data than Sanger sequencing in shorter timeframes, making it highly efficient for whole mitochondrial genome analysis. Its accuracy and cost-effectiveness allow for the simultaneous sequencing of multiple mtDNA genomes, providing deeper insights into genetic diversity and phylogenetic relationships. NGS is particularly valuable for degraded and ancient samples, as demonstrated in paleogenomic research [64]. Applying NGS to ancient horse remains could clarify missing lineage connections, enhance phylogenetic accuracy, and refine haplogroup classifications, ultimately providing a more complete picture of equine maternal evolution.

Beyond evolutionary and conservation implications, integrating haplotype and haplogroup classifications with known performance outcomes in different sports could reveal significant correlations between mtDNA haplogroups and performance traits. This could provide valuable insights for breeding and training programs, optimizing selection strategies based on genetic predispositions. Notably, only two of the extracted articles attempted to explore this relationship [65,66].

Moving forward, advancing equine mtDNA research requires an integrated approach that combines whole mitochondrial genome sequencing, Y-chromosome and autosomal markers, ancient DNA analysis, and standardized haplogroup classification. These efforts will be essential for improving phylogenetic resolution, enhancing conservation strategies, and uncovering the full extent of equine genetic diversity.”

Comment 2/ The authors focused only on mtDNA D-loop variation, without discussing the Cytochrome b gene, COX1 gene, and other coding regions of mtDNA. In my opinion, these regions should also be included in the review to provide a comprehensive overview of the manuscript.

Response: 

  • We focused on the D-loop because most of the articles use it, only two talked about Cyt b and even they used it in conjunction with the D-loop. We added this paragraph to the discussion (476-483):

“Most studies focused on the D-loop, which is widely used due to its high mutation rate and informativeness for maternal lineage studies. However, only two studies analyzed cytochrome b, and both of these also included the D-loop [61,62]. Sziszkosz et al. [62] compared the two markers and found that some haplotypes were exclusive to cytochrome b, others to the D-loop, while some appeared only when combining both markers. These findings suggest that using multiple mitochondrial markers provides more precise results than analyzing the D-loop alone, as relying solely on one region may overlook critical genetic variations.”

Comment 3/ In my opinion, instead of discussing the review collection methodology, the authors should focus on how the review would contribute to future research, especially in the areas of phylogenetic differentiation or sequence variations among horse breeds such as Arabian, Thoroughbred, and Mongolian. The review should also highlight its application in domestication events, migration patterns, and breed evolution. Although mtDNA is primarily used to study the maternal lineages of different horse breeds, the authors have missed providing information on predominant research works related to population structure analysis and the phylogenetic evolutionary relationship between horse breeds.

Response: 

  • We Extracted and added information about the application of mtDNA analysis in each study to Table S1 to better categorize how mtDNA has been used in research.
  • We Presented the statistics and key findings in the Results section (181-213) :

“The included studies demonstrated various applications of mtDNA analysis (Ta-ble S1). The majority (85.5%) used mtDNA to analyze the population structure of a given group. Almost all studies (97.3%) applied mtDNA to compare phylogenetic rela-tionships, either between specific populations or in relation to different haplogroup systems, as well as to determine the origin of a breed. Additionally, 48.6% of the studies discussed conservation strategies based on mtDNA findings, while 11.8% used mtDNA to verify studbooks and pedigree records of their horses (Fig 2).

Figure 2. Histogram of mtDNA applications.

The key findings from these studies converge on several themes, with some variations. Most research focuses on the genetic diversity of specific horse breeds, often confirming historical records [16,17], hypothesizing about their origins [18–20], or exploring their genetic relationships with other breeds [21–24]. For instance, Giontella et al. [19] supports the hypothesis that both humans and Giara horses migrated from the Eastern Mediterranean during the first millennium BCE.

Beyond origins, several studies document the loss of genetic diversity over time, particularly the decline of specific haplotypes [25–27]. In Chinese indigenous horses, haplogroups B, F, and G have decreased in frequency in recent years [25]. Similarly, Spanish endangered breeds have lost maternal lines [27]. This decline highlights the connection between genetic diversity loss and breed extinction, reinforcing the need to integrate mtDNA with autosomal and paternal markers for more effective conservation strategies [25].

Selection pressures have also shaped horse populations. Novoa-Bravo et al. [28] demonstrated microevolutionary changes leading to the divergence of two breeds due to selective breeding. Winton et al. [29] emphasized the role of subpopulation analysis in understanding the genetic impact of past management practices and guiding future conservation efforts.

Finally, in pedigree verification, some studies confirm the reliability of studbooks [30]. Bower et al. [31] reports that maternal sub-lineage records for Thoroughbred racehorses are 92% accurate, compared to only 60% for general maternal lineage records. However, other studies reveal inconsistencies in pedigree documentation [27,32,33], underscoring the need for genetic tools to enhance accuracy in pedigree management.”

Comment 4/ The review may be modified to focus on the current challenges, applications of mtDNA research, future research perspectives, and limitations of mtDNA use in research work.

Response: 

  • We detailed these more in discussion, limitations and perspectives and in the conclusion.

Comment 5/ Lines 190-194: The review by Gurău et al. [15] discusses non-invasive DNA isolation from goat samples, which is not directly relevant to the manuscript. It would be more appropriate to replace this with studies focusing on non-invasive DNA isolation techniques in horses, such as hair follicles or buccal swabs, to better align with the manuscript's scope.

Response: 

  • Since Hair was used in nearly 50% of the studies we analysed, we wanted to compare it to blood, but we couldn’t any article that directly compare blood and hair for DNA extraction in horses. We kept Gurau et al but added one study comparing Feces and oral mucosal swabs to blood. (444-464)

Blood was the most commonly used sample because it provides sufficient high-quality DNA for genetic analysis, though collection can sometimes be challenging. Non-invasive sampling methods —including faeces, hair, saliva—are easier to collect and less invasive than blood. As a result, they are increasingly used to address animal welfare concerns [55]. In a systematic literature review, Zemanova et al.[55] reported a steady increase in the use of non-invasive methods in wildlife research.
Comparative studies have evaluated DNA isolation efficiency across different sample types. Gurău et al.
[56] compared DNA extraction from blood and hair follicle samples in goats, finding that while whole blood yielded slightly higher DNA quantities, both sample types provided sufficient DNA for amplifying 1,200 bp fragments. Similarly, Muko et al. [57] assessed DNA yield and quality from oral mucosa swabs and faeces in horses, comparing them with DNA from blood. Although non-invasive samples provided sufficient DNA for PCR analysis, contamination from microorganisms and feed remained a concern.

In a review of 113 studies, Zemanova et al. [55] found that 94% of studies reported equivalent or superior performance of non-invasive genetic assessments compared to invasive sampling. Non-invasive techniques can also be more cost-effective and time-efficient. Thus, adopting non-invasive extraction methods could simplify DNA collection while maintaining comparable DNA quality. This approach may also achieve similar sequencing performance for the entire D-loop region of mtDNA, which is approximately 1,192 bp long [17]. “

Round 2

Reviewer 1 Report

Comments and Suggestions for Authors

Dear Editor and Authors,

I have carefully reviewed the revised manuscript and am pleased to note that the requested improvements have been successfully implemented. I have no further comments or suggestions, and I accept the manuscript for publication in its current form.

Best regards,

Reviewer 2 Report

Comments and Suggestions for Authors

The authors addressed my comments, no further issues, except for Table S1 is still unaccessible, but Google reference works fine, thank you.

Reviewer 3 Report

Comments and Suggestions for Authors

The manuscript has improved with the given corrections, and the article may be accepted for publication.